# Design, Synthesis, and Antibacterial Screening of Some Novel Heteroaryl-Based Ciprofloxacin Derivatives as DNA Gyrase and Topoisomerase IV Inhibitors

**DOI:** 10.3390/ph14050399

**Published:** 2021-04-22

**Authors:** Lamya H. Al-Wahaibi, Amer A. Amer, Adel A. Marzouk, Hesham A. M. Gomaa, Bahaa G. M. Youssif, Antar A. Abdelhamid

**Affiliations:** 1Department of Chemistry, College of Sciences, Princess Nourah Bint Abdulrahman University, Riyadh P.O. Box 84428, Saudi Arabia; lhalwahaibi@pnu.edu.sa; 2Department of Chemistry, Faculty of Science, Sohag University, Sohag 82524, Egypt; amer_chem@yahoo.com (A.A.A.); drantar25@yahoo.com (A.A.A.); 3Department of Pharmaceutical Chemistry, Faculty of Pharmacy, Al-Azhar University, Assiut 71524, Egypt; adelmarzouk@azhar.edu.eg; 4Pharmacology Department, College of Pharmacy, Jouf University, Aljouf 72341, Saudi Arabia; hasoliman@ju.edu.sa; 5Pharmaceutical Organic Chemistry Department, Faculty of Pharmacy, Assiut University, Assiut 71526, Egypt; 6Chemistry Department, Faculty of Science, Albaha University, Albaha P.O. Box 1988, Saudi Arabia

**Keywords:** ciprofloxacin, heteroaryl, antibacterial, gyrase, topoisomerase IV

## Abstract

A novel series of ciprofloxacin hybrids comprising various heterocycle derivatives has been synthesized and structurally elucidated using ^1^H NMR, ^13^C NMR, and elementary analyses. Using ciprofloxacin as a reference, compounds **1–21** were screened in vitro against Gram-positive bacterial strains such as *Staphylococcus aureus* and *Bacillus subtilis* and Gram-negative strains such as *Escherichia coli* and *Pseudomonas aeruginosa*. As a result, many of the compounds examined had antibacterial activity equivalent to ciprofloxacin against test bacteria. Compounds **2–6**, oxadiazole derivatives, were found to have antibacterial activity that was 88 to 120% that of ciprofloxacin against Gram-positive and Gram-negative bacteria. The findings showed that none of the compounds tested had antifungal activity against *Aspergillus flavus*, but did have poor activity against *Candida albicans*, ranging from 23% to 33% of fluconazole, with compound **3** being the most active (33% of fluconazole). The most potent compounds, **3**, **4**, **5**, and **6**, displayed an IC_50_ of 86, 42, 92, and 180 nM against *E. coli* DNA gyrase, respectively (novobiocin, IC_50_ = 170 nM). Compounds **4**, **5**, and **6** showed IC_50_ values (1.47, 6.80, and 8.92 µM, respectively) against *E. coli* topo IV in comparison to novobiocin (IC_50_ = 11 µM).

## 1. Introduction

Bacterial infection remains a significant threat to human life due to its increasing resistance to conventional antibiotics, which is a growing public health concern. As a result, there is a critical need to create new antimicrobial agents with potent anti-drug-resistant microorganism activity [1]. That is why antimicrobial agent investigations are so critical and should always be up to date.

Due to their excellent efficacy, bioavailability, and relatively low toxic and adverse effects, fluoroquinolone (FQ) antibiotics have been one of the most commonly used groups of antibiotics in recent years, (Figure 1). Fluoroquinolones (FQs) are antibiotics that have the ability to treat a range of bacterial infections [2,3]. FQs are inhibitors of *S. aureus* multidrug efflux pumps [4], lower and upper respiratory infections [5], prostatitis, and urinary tract infections (UTI) [6]. Fluoroquinolone antibiotics co-alter the bacterial DNA gyrase and topoisomerase IV enzymes in a hybrid enzyme–DNA complex [7]. Such a change in bacterial enzyme performance inhibits the desired bacterial growth DNA synthesis. FQ drugs have long been known as a favored structural framework for the development of new commercially available drugs [8]. Changes in the elementary structure of FQs are thought to enhance drug interaction with the target enzyme, which may improve pharmacokinetic properties. This type of structural functioning of the FQs may provide stronger drugs for future generations. Substitution at position 7 of the FQ core has a significant impact on solubility, bioavailability, and antimicrobial activities [9]. With the introduction of various C-7 moieties, researchers have focused on improving their biological range. As a result, different moieties were used to examine its biological profile with a piperazinyl ring, substituted piperazine moiety, heterocyclic ring (especially five or six members). Ciprofloxacin is one of the most potent second-generation fluoroquinolones (FQs). It offers potential anti-infective therapy for a wide range of bacteria, Gram-positive, and Gram-negative (Figure 1) [10,11]. For 12 medical treatments, including veterinary uses, the Food and Drug Administration (FDA) approved ciprofloxacin (CP, see Figure 1) which exhibits antibacterial activity with minimal side effects and good pharmacokinetic properties. In addition, ciprofloxacin has a wide range of biological profiles and has been used to examine its antimalarial, anti-fungal, anti-tumor, and antibacterial properties in different areas of medical research [12,13].

Ciprofloxacin is currently used to treat a number of Gram-positive and Gram-negative bacterial infections in clinical practice. However, because of the emergency and widespread of drug-resistant bacteria, ciprofloxacin is becoming increasingly ineffective. As a result, novel antibacterial agents are urgently needed [14].

To combat resistance, the production of novel ciprofloxacin derivatives that are effective against both drug-susceptible and drug-resistant pathogens is crucial. Many ciprofloxacin derivatives have been designed and synthesized with excellent in vitro and in vivo potency against both drug-sensitive and drug-resistant species, including fluoroquinolone-resistant, multidrug-resistant pathogens [15,16,17,18,19].

With MICs ranging from 0.28 to 15.8 µM/mL, the 1,3,4-oxadiazole ciprofloxacin hybrid **I** (Figure 2) was >41 times more potent than ampicillin (MIC: 10–128 mg/mL) against most of the pathogens tested [15].

The antibacterial and antifungal activities of the thiadiazole ciprofloxacin hybrids **II** (Figure 2) were only mild to moderate [16].

Agarwal et al. tested a sequence of bis-1,2,3-triazole-ciprofloxacin hybrids **III** (Figure 2) in vitro against a panel of clinically important bacteria [17]. A significant portion of the hybrids showed increased activity against both Gram-positive and Gram-negative bacteria relative to ciprofloxacin, and antibacterial activity appears to be related to the nature and position of substituents, as well as their isomeric effects on phenyl rings. Furthermore, the compounds’ low toxicity profile suggests that they may be useful antibiotics in the future [17].

Demirbas et al. found that the 1,2,4-triazole-5(4*H*)-one/thione ciprofloxacin hybrids **IV** and **V** (Figure 2) with substituted piperazine at the C-3 position of the triazole moiety had promising in vitro activity against both Gram-positive and Gram-negative pathogens with MIC 0.24 mg/mL, which was far more potent than Ampicillin (MIC: 3.9–250 mg/mL) [18,19].

Continuing our quest to find a compound with improved antimicrobial properties [20,21,22,23,24,25,26,27], the current study describes the synthesis and structure elucidation of Fluoroquinolone(ciprofloxacin)-based hybrids containing various heterocycle derivatives, as well as antimicrobial activity evaluation using various strains of Gram-positive (*S. aureus* and *B. subtilis*), Gram-negative (*E. coli* and *P. aeruginosa*), and fungi (*A. flavus* and *C. albicans*). In addition, the inhibitory activity of the most active compounds against *E. coli* DNA gyrase and topoisomerase IV has been identified as a potential molecular target.

## 2. Results and Discussion

### 2.1. Chemistry

The synthetic strategy towards the synthesis of target compounds **1–21** is outlined in Scheme 1. The intermediates **VIa–f [28]**, **VIIa–f [29]**, **VIIIa–c [30]**, and **IXa–d [31]** were synthesized, as stated previously, and their structures were confirmed by comparing their physical constants and spectral data to those previously recorded. As shown in Scheme 1, ciprofloxacin undergoes a Mannich reaction with heterocycles **I**–**VI** and formaldehyde in refluxing ethanol to yield the target compounds **1**–**21** in yields ranging from 79% to 97%. The IR spectra of compounds **1**–**21** showed a stretching band at 3381–3310 cm^−1^ related to (OH), a medium stretching band at 3062–3004 of CH aromatic and strong stretching band at 1731–1701 cm^−1^ related to (C=O), which are consistent with the proposed structure. The ^1^H NMR spectra of **1**–**21** revealed the appearance of a methylene signal at 5.19–5.16 (s, 2H, N-CH_2_-N), two sets of triplets at 3.39–3.27 and 3.07–3.03 ppm, which is indicative of piperazinyl protons, three signals at 3.84–3.80 (m, 1H), 1.35–1.30 (q, 2H), and 1.18–1.14 (q, 2H) ppm attributed to cyclopropyl protons, and broad singlet at 15.13–15.05 ppm of the COOH group. Moreover, the ^1^H NMR spectra of **16–19** revealed the appearance of singlet signal at 8.64 ppm of olefinic CH. The ^13^C NMR spectra of **1–21** showed the characteristic methyl carbon (N-C-N) at 70 ppm and the (C=O) at 166–160 and 173–178 ppm. At their predicted chemical shifts were the olefin and aromatic carbons (Appendix A). The purity of **1–21** has been confirmed using elementary analysis, and the results fit the products’ molecular formula.

### 2.2. Biology

#### 2.2.1. Antimicrobial Sensitivity Test

An updated Kirby-Bauer disc diffusion method was used to assess the antimicrobial activity of the tested samples [32,33,34,35]. Table 1 presents the results of the preliminary antimicrobial testing of final compounds. Using ciprofloxacin as a reference drug, synthetic compounds **1–21** were tested in vitro for antibacterial activity against *S. aureus* and *B. subtilis* as Gram-positive strains and *E. coli* and *P. aeruginosa* as Gram-negative strains. As a result, the majority of newly synthesized compounds demonstrated promising antibacterial activity comparable to ciprofloxacin against test species (Table 1). The oxadiazole derivatives, compounds **2–6** were found to exhibit pronounced antibacterial activity, which ranged from 88% to 120% that of ciprofloxacin against both Gram-positive and Gram-negative strains. It is worth mentioning that compound **6** showed superior activity (120%) against *S. aureus* to that of ciprofloxacin. Oxadiazoles **4** and **5** showed equipotent activity to ciprofloxacin against *S. aureus* and *E. coli*. The thiazolidine derivative **16** had a ciprofloxacin-like activity of 93% against *B. subtilis*, *E. coli*, and *P. aeruginosa*, but only 85% activity against *S. aureus*, Table 1. Compounds **1**, **7–10**, and **17–21** showed moderate antibacterial activity which ranged from 70% to 83% of ciprofloxacin against both Gram-positive and Gram-negative strains.

According to the previous findings, the inclusion of oxadiazole and thiadiazole moiety in a compound confers the highest efficacy.

Furthermore, using fluconazole as a reference drug, **1**–**21** were tested in vitro for antifungal activity against *A. flavus* and *C. albicans* (Table 1). The findings showed that the tested compounds had no antifungal activity against *A. flavus* but had weak antifungal activity against *C. albicans*, ranging from 23% to 33% of fluconazole, with compound **3** being the most active (33% of fluconazole).

#### 2.2.2. Minimum Inhibitory Concentration Test

A two-fold serial dilution method was used to assess the antimicrobial activity of the most active components, oxadiazole-based hybrids, **2–6 [36]**. Table 2, using the reference drug ciprofloxacin, was presented with the MIC_s_ (minimal inhibitory concentrations) of these compounds against tested bacteria. Some new compounds have demonstrated good antimicrobial inhibitory activities for Gram-positive and Gram-negative strains. As shown in Table 2, compounds **4**, **5**, and **6** were the most active and effective against three bacterial strains, in which compound **4** with MIC values of 0.035, 0.062, 0.062 µg/mL against *S. aureus*, *E. coli* and *P. aureoginosa*, compound **5** with MIC values of 0.035, 0.062, 0.125 µg/mL against *S. aureus*, *E. coli* and *P. aureoginosa*, and compound **6** with MIC values of 0.031, 0.125, 0.125 µg/mL against *S. aureus*, *E. coli* and *P. aureoginosa*, respectively. In comparison to ciprofloxacin (MIC of 0.030 g/mL), compound **2** showed the next best activity against *S. aureus* strains with a MIC value of 0.062 g/mL. Interestingly, all the compounds examined only had a minor inhibitory effect on *B. subtilis*, a Gram-positive organism. The nature of aromatic substitution in the oxadiazole moiety tends to be correlated with higher antibacterial effects and the activity increased with (Ar) in the order of 2-pyridyl ≥ 3-pyridyl > 4-pyridyl > 2-Cl-Ph > 2-naphthyl.

#### 2.2.3. Inhibitory Activity Against *E. coli* DNA Gyrase and Topoisomerase IV 

*E. coli* DNA gyrase assay [37] was performed to evaluate the inhibitory potency of oxadiazole-based derivatives **2–6** against *E. coli* DNA gyrase and the results are included in Table 3. Results are presented as residual activities (RAs) of the enzyme at 1 µM of compounds or as IC_50_ values for compounds with RA <50% (Table 3). The results of the antimicrobial activity study are complemented by the results from this assay. Investigated compounds **3**–**6** exhibited inhibition of *E. coli* DNA with IC_50_ ranging from 42 to 180 nM relative to reference novobiocin (IC_50_ = 170 nM). Based on the data provided, compounds **3**, **4**, and **5** were found to be the most active and their inhibitory activities of *E. coli* DNA gyrase assay (IC_50_ = 86 ± 9, 42 ± 7, and 92 ± 9, respectively) were superior to positive control novobiocin.

Compounds **2–6** were further evaluated against *E. coli* topoisomerase IV [37], as shown in Table 3. Compounds **4**, **5,** and **6**, which were among the most potent inhibitors of *E. coli* gyrase also displayed promising results on topoisomerase IV (Table 3). Compounds **4**, **5**, and **6** had IC_50_ values = 1.47, 6.80, and 8.92 µM, respectively, which are much lower (more potent) than that for novobiocin (IC_50_ = 11 µM). From these findings, both **4** and **5**, after optimization, appear to be promising dual target inhibitors against DNA gyrase and topoisomerase IV.

#### 2.2.4. Cell Viability Assay

A human mammary gland epithelial cell line (MCF-10A) was used to conduct a cell viability assay [38]. Compounds **2–6** were incubated with MCF-10A cells for four days, and the viability of the cells was determined using the 3-(4,5-dimethylthiazol-2-yl)-2,5-diphenyltetrazolium bromide (MTT) assay [39]. All compounds had no cytotoxic effects, and the viability of the cells was greater than 85% for most of the compounds examined at 50 M, as shown in Table 4.

### 2.3. Drug Likeness Profile

Absorption, delivery, metabolism, and excretion (ADME) testing became popular early in drug development programs, with computer models serving as feasible alternatives to experiments. The Swiss ADME website was used to predict the drug likeness profile of the studied compounds 1–21 [40,41]. Appendix A display the results of the drug likeness profile of these compounds. Many compounds, such as **1**, **13**, **14**, **16**, **17**, **20**, and **21**, were expected to have high oral absorption. However, due to the high molecular weight (522) and molar refractivity (143), the others predicted poor oral absorption. Some of the tested compounds as 20 and 21 showed no violation to Lipinski (Pfizer) filters, except for one violation for compounds 1–8 and 11–19. In addition two violation for compounds 9 and 10 (molecular weight >500 [42]) and compound 21 showed no violations to Ghose in addition to two violations for 1–6, three violations for 7, 8, 11, 12, 13, 14 and four violations for 9 and 10 due to high WLOGP and molecular weight [43], no violation to Veber (GSK) for all compounds except one violation for 4, 5, 6, 14, 15, 19 [44], no violation to Egan (Pharmacia) for all compounds except one violation for 4, 5, 6, 14, 15, 18, 19 [45] and no violation to Muegge (Bayer) except one violation for compounds 1, 8, 13, 14, and two violations for 9, 10 [46] filters. The compounds were free from alerts for Pan Assay Interfering substances (PAINS) [47]. Total polar surface area (TPSA) values for most compounds are 103.16–148.69, as shown in Table 1, Table 2 and Table 3. This consists of “good GIT absorption”. There is a correlation between the molecular weight of compounds and their activity. In addition to low rigidity, this pattern highlighted low molecular weight is favorable. Lipophilicity, together with the molecular weight and the number of hydrogen bond donors and the number of hydrogen acceptors shown by these compounds, plays the role of five (see Appendix A).

## 3. Materials and Methods

### 3.1. Chemistry

General Details: See Appendix B

#### 3.1.1. General Procedure for the Synthesis of Compounds **1**–**21**

To a mixture of ciprofloxacin HCl (1.28 g, 0.003 mol), formaldehyde (0.2 g, 0.007 mol) and different heterocyclic compounds (0.003 mol) were dissolved in 10 mL of ethanol. The reaction mixture was stirred with reflux for 30 min. The reaction mixture was allowed to cool at room temperature, the separated solid was filtered off, washed with water, and crystallized from methanol.

##### 1-Cyclopropyl-6-fluoro-4-oxo-7-[4-(5-phenyl-2-thioxo[1,3,4]oxadiazol-3-yl-methyl)-piperazin-1-yl]-1,4-dihydro-quinoline-3-carboxylic acid (**1**)

Yield (94%); m.p. 228 °C; IR: 3350 (OH), 3049 (CH aromatic), 2987, 2946 (CH-aliphatic), 1710 (C=O), 1615 (C=N); ^1^H NMR: δ 15.13 (brs. OH, COOH), 8.65–7.58 (m, 7H, aromatic), 5.16 (s, 2H, N-CH_2_-N), 3.83–3.80 (m, J = 2.44 Hz, 1H, CH_2_-CH-CH_2_ cyclopropyl-H), 3.38 (t, 4H, piperazinyl-H), 3.04 (t, 4H, piperazinyl-H), 1.34–1.32 (q, J = 3.54 Hz, 2H, CH_2_-CH-CH_2_ cyclopropyl-H), 1.18–1.15 (q, J = 4.08 Hz, 2H, CH_2_-CH-CH_2_cyclopropyl-H); ^13^C NMR: δ 178.82., 166.40, 148.28. 145.45, 139.64, 129.88, 111.52, 107.27, 106.96, 70.24, 51.05 49.82, 49.70, 36.27, 8.02. C_26_H_24_FN_5_O_5_S: C, 59.87; H, 4.64; N, 13.43, Found: C, 59.81; H, 4.62; N, 13.34.

##### 7-{4-[5-(2-Chloro-phenyl)-2-thioxo-[1,3,4]oxadiazol-3-ylmethyl]-piperazin-1-yl}-1-cyclopropyl-6-fluoro-4-oxo-1,4-dihydro-quinoline-3-carboxylic acid (*2*)

Yield (85%); m.p. 226 °C; IR: 3354 (OH), 3062 (CH aromatic), 2999, 2914 (CH-aliphatic), 1719 (C=O), 1614 (C=N); ^1^H NMR: δ 15.11 (brs. OH, COOH), 8.65–7.65 (m, 7H, aromatic), 5.19 (s, 2H, N-CH_2_-N), 3.84–3.80 (m, J = 3.52 Hz, 1H, CH_2_-CH-CH_2_ cyclopropyl-H), 3.39 (t, 4H, piperazinyl-H), 3.03 (t, 4H, piperazinyl-H), 1.35–1.30 (q, J = 5.44 Hz, 2H, CH_2_-CH-CH_2_ cyclopropyl-H), 1.18–1.15 (q, J = 5.64 Hz, 2H, CH_2_-CH-CH_2_cyclopropyl-H); ^13^C NMR: δ 173.59., 168.75, 166.45, 131.73, 128.38, 111.55, 111.30, 110.09, 109.37, 108.30, 107.17, 70.26, 49.84, 49.62, 36.33, 8.03. C_26_H_23_ClFN_5_O_5_S: C, 56.16; H, 4.17; N, 12.60, Found: C, 56.08; H, 4.02; N, 12.34.

##### 1-Cyclopropyl-6-fluoro-7-[4-(5-naphthalen-2-yl-2-thioxo-[1,3,4]oxadiazol-3-yl methyl)-piperazin-1-yl]-4-oxo-1,4-dihydro-quinoline-3-carboxylic acid (**3**)

Yield (96%); m.p. 244 °C; IR: 3367 (OH), 3019 (CH aromatic), 2961, 2957 (CH-aliphatic), 1708 (C=O), 1609 (C=N); ^1^H NMR: δ 15.11 (brs. OH, COOH), 8.66–7.55 (m, 10H, aromatic), 5.19 (s, 2H, N-CH2-N), 3.83–3.79 (m, J = 5.51 Hz, 1H, CH_2_-CH-CH_2_ cyclopropyl-H), 3.39 (t, 4H, piperazinyl-H), 3.07 (t, 4H, piperazinyl-H), 1.35–1.30 (q, 4H, J = 5.69 Hz, CH_2_-CH-CH_2_ cyclopropyl-H); ^13^C NMR: δ 176.74, 166.27, 148.44, 139.39 122.25, 111.72, 107.27, 49.91, 49.79, 36.29, 8.03. C_30_H_26_FN_5_O_5_S: C, 63.03; H, 4.58; N, 12.25, Found: C, 62.88; H, 4.62; N, 12.01.

##### 1-Cyclopropyl-6-fluoro-4-oxo-7-[4-(5-pyridin-2-yl-2-thioxo-[1,3,4]oxadiazol-3-yl-methyl)-piperazin-1-yl]-1,4-dihydro-quinoline-3-carboxylic acid (**4**)

Yield (89%); m.p. 218 °C; IR: 3369 (OH), 3049 (CH aromatic), 2967, 2924 (CH-aliphatic), 1721 (C=O), 1619 (C=N); ^1^H NMR: δ 15.10 (brs. OH, COOH), 9.07–7.55 (m, 7H, aromatic), 5.17 (s, 2H, N-CH_2_-N), 3.80 (m, J = 2.8 Hz, 1H, CH_2_-CH-CH_2_ cyclopropyl-H), 3.27 (t, 4H, piperazinyl-H), 3.04 (t, 4H, piperazinyl-H), 1.30–1.35 (q, J = 5.20 Hz, 2H, CH_2_-CH-CH_2_ cyclopropyl-H), 1.14–1.18 (q, J = 6.92 Hz, 2H, CH_2_-CH-CH_2_cyclopropyl-H); ^13^C NMR: δ 176.80., 166.34, 148.40. 142.31, 139.62, 124.25, 119.74, 111.25, 107.25, 107.00, 70.32, 49.84, 49.69, 36.28, 18.95 8.02. C_25_H_24_FN_6_O_5_S: C, 57.46; H, 4.44; N, 16.08, Found: C, 57.29; H, 4.31; N, 16.01.

##### 1-Cyclopropyl-6-fluoro-4-oxo-7-[4-(5-pyridin-3-yl-2-thioxo-[1,3,4]oxadiazol-3-yl-methyl)-piperazin-1-yl]-1,4-dihydro-quinoline-3-carboxylic acid (**5**)

Yield (92%); m.p. 230 °C; IR: 3310 (OH), 3019 (CH aromatic), 2984, 2902 (CH-aliphatic), 1707 (C=O), 1609 (C=N); ^1^H NMR: δ 15.11 (brs. OH, COOH), 8.77–7.55 (m, 1H, aromatic), 5.18 (s, 2H, N-CH_2_-N), 3.82–3.80 (m, J = 1.80 Hz, 1H, CH_2_-CH-CH_2_ cyclopropyl-H), 3.27(t, 4H, piperazinyl-H), 3.04 (t, 4H, piperazinyl-H), 1.33–1.32 (q, J = 1.96 Hz, 2H, CH_2_-CH-CH_2_ cyclopropyl-H), 1.17–1.16 (q, J = 1.76 Hz, 2H, CH_2_-CH-CH_2_cyclopropyl-H); ^13^C NMR: δ 176.73, 166.41, 154.62, 152.14, 150.78, 148.37, 145.40, 139.55, 138.32, 127.27, 123.06, 118.96, 111.44, 111.22, 107.13, 106.96, 70.30, 49.79, 49.65, 43.21, 36.28, 18.98, 8.02. C_25_H_24_FN_6_O_5_S: C, 57.46; H, 4.44; N, 16.08, Found: C, 57.33; H, 4.40; N, 16.41.

##### 1-Cyclopropyl-6-fluoro-4-oxo-7-[4-(5-pyridin-4-yl-2-thioxo-[1,3,4]oxadiazol-3-yl-methyl)-piperazin-1-yl]-1,4-dihydro-quinoline-3-carboxylic acid (**6**)

Yield (93%); m.p. 208 °C; IR: 3351 (OH), 3004 (CH aromatic), 2995, 2913 (CH-aliphatic), 1731 (C=O), 1607 (C=N); ^1^H NMR: δ 15.05 (brs. OH, COOH), 8.81–7.57 (m, 7H aromatic), 5.17 (s, 2H, N-CH_2_-N), 3.82 (m, J = 3 Hz, 1H, CH_2_-CH-CH_2_ cyclopropyl-H), 3.22(t, 4H, piperazinyl-H), 3.05 (t, 4H, piperazinyl-H), 1.35–1.33 (q, J = 4.56 Hz, 2H, CH_2_-CH-CH_2_ cyclopropyl-H), 1.19–1.17 (q, J = 3.04 Hz, 2H, CH_2_-CH-CH_2_cyclopropyl-H); ^13^C NMR: δ 178.81., 166.31, 157.70, 154.65, 151.34, 140.37, 139.62, 130.27, 120.07, 119.08, 111.51, 111.28, 107.29, 106.99, 70.28, 49.84, 49.68, 36.27, 8.03. C_25_H_24_FN_6_O_5_S: C, 57.46; H, 4.44; N, 16.08, Found: C, 57.14; H, 4.39; N, 16.15.

##### 1-Cyclopropyl-7-[4-(3,4-diphenyl-5-thioxo-4,5-dihydro-[1,2,4]triazol-1-ylmethyl)-piperazin-1-yl]-6-fluoro-4-oxo-1,4-dihydro-quinoline-3-carboxylic acid (**7**)

Yield (93%); m.p. 288 °C; IR: 3381 (OH), 3014 (CH aromatic), 2961, 2943, 2906 (CH-aliphatic), 1707 (C=O), 1598 (C=N); ^1^H NMR: δ 15.12 (brs. OH, COOH), 8.66–7.36 (m, 13H, aromatic), 5.30 (s, 2H, N-CH_2_-N), 3.80–3.85 (m, J = 7.64 Hz, 1H, CH_2_-CH-CH_2_ cyclopropyl-H), 3.3.41 (t, 4H, piperazinyl-H), 3.09 (t, 4H, piperazinyl-H), 1.31–1.36 (q, J = 7.45 Hz, 2H, CH_2_-CH-CH_2_ cyclopropyl-H), 1.19–1.15 (q, J = 6.76 Hz, 2H, CH_2_-CH-CH_2_cyclopropyl-H); ^13^C NMR: δ 177.32, 173.40, 172.02, 170.78, 170.66, 150.17, 149.38, 148.54, 143.74, 136.56, 135.68, 135.68, 134.41, 129.07, 128.89, 58.17, 52.68, 50.08, 50.08, 10.64. C_32_H_29_FN_6_O_3_S: C, 64.41; H, 4.90; N, 14.08, Found: C, 64.22; H, 4.81; N, 14.02.

##### 7-{4-[3-(2-Chloro-phenyl)-4-phenyl-5-thioxo-4,5-dihydro-[1,2,4]triazol-1-ylmethyl]-piperazin-1-yl}-1-cyclopropyl-6-fluoro-4-oxo-1,4-dihydro-quinoline-3-carboxylic acid (**8**)

Yield (96%); m.p. 214 °C; IR: 3332 (OH), 3056 (CH aromatic), 2984, 2917 (CH-aliphatic), 1701 (C=O), 1614 (C=N); ^1^H NMR: δ 15.13 (brs. OH, COOH), 8.67–7.33 (m, 12H, aromatic), 5.33 (s, 2H, N-CH_2_-N), 3.84–3.83 (m, J = 3.00 Hz, 1H, CH_2_-CH-CH_2_ cyclopropyl-H), 3.42 (t, 4H, piperazinyl-H), 3.07 (t, 4H, piperazinyl-H), 1.34–1.19 (q, J = 1.40 Hz, 4H, CH_2_-CH-CH_2_ cyclopropyl-H); ^13^C NMR: δ 176.84, 169.33, 166.44, 148.51, 147.50, 145.60, 134.35, 133.45, 129.55, 129.45, 128.57, 127.04, 111.57, 107.19, 107.04, 69.24, 50.06, 49.90, 36.33, 8.08. C_32_H_28_ClFN_6_O_3_S: C, 60.90; H, 4.47; N, 13.32, Found: C, 60.59; H, 4.28; N, 13.39.

##### 1-Cyclopropyl-6-fluoro-7-[4-(3-naphthalen-2-yl-4-phenyl-5-thioxo-4,5-dihydro-[1,2,4]triazol-1-ylmethyl)-piperazin-1-yl]-4-oxo-1,4-dihydro-quinoline-3-carboxylic acid (**9**)

Yield (97 %); m.p. 262 °C; IR: 3365 (OH), 3041 (CH aromatic), 2964, 2921 (CH-aliphatic), 1714 (C=O), 1612 (C=N); ^1^H NMR: δ 15.14 (brs. OH, COOH), 8.65–7.43 (m, 15H, aromatic), 5.34 (s, 2H, N-CH_2_-N), 3.82 (m, CH_2_-CH-CH_2_ cyclopropyl-H), 3.42 (t, 4H, piperazinyl-H), 3.12 (t, 4H, piperazinyl-H), 1.33–1.31 (q, J = 0.74 Hz, 2H, CH_2_-CH-CH_2_ cyclopropyl-H), 1.17–1.16 (q, J = 1.52 Hz, 2H, CH_2_-CH-CH_2_cyclopropyl-H); ^13^C NMR: δ 176.81, 170.25, 166.34, 149.70, 148.41, 145.70, 139.62, 135.80, 133.74, 132.42, 130.04, 129.86, 129.28, 129.12, 128.63, 128.25, 127.56, 123.18, 111.53, 111.30, 107.25, 106.97, 69.41, 50.16, 49.96, 36.29, 8.04. C_36_H_31_FN_6_O_3_S: C, 66.86; H, 4.83; N, 12.99, Found: C, 66.49; H, 4.69; N, 12.58.

##### 1-Cyclopropyl-6-fluoro-4-oxo-7-[4-(4-phenyl-3-pyridin-2-yl-5-thioxo-4,5-dihydro-[1,2,4]triazol-1-ylmethyl)-piperazin-1-yl]-1,4-dihydro-quinoline-3-carboxylic acid (**10**)

Yield (96%); m.p. 270 °C; IR: 3371 (OH), 3095 (CH aromatic), 2996, 2915 (CH-aliphatic), 1704 (C=O), 1611 (C=N); ^1^H NMR: δ 15.12 (brs. OH, COOH), 8.63–7.34 (m, 12H, aromatic), 5.33 (s, 2H, N-CH_2_-N), 3.82–3.79 (m, J = 3.88 Hz, 1H, CH_2_-CH-CH_2_ cyclopropyl-H), 3.40(t, 4H, piperazinyl-H), 3.08 (t, 4H, piperazinyl-H), 1.35–1.30 (q, J = 6.49 Hz, 2H, CH_2_-CH-CH_2_ cyclopropyl-H), 1.18–1.15 (q, J = 6.44 Hz, 2H, CH_2_-CH-CH_2_cyclopropyl-H); ^13^C NMR: δ 176.80, 170.87, 166.33, 154.88, 152.20, 149.73, 148.67, 148.31, 145.60, 145.28, 139.82, 137.72, 136.05, 129.41, 129.24, 128.87, 125.41, 124.65, 119.08, 111.52, 111.25, 107.33, 106.89, 69.45, 50.16 49.93, 49.53, 36.26, 8.03. C_31_H_28_FN_7_O_3_S: C, 62.30; H, 4.72; N, 16.41, Found: C, 62.04; H, 4.61; N, 16.21.

##### 1-Cyclopropyl-6-fluoro-4-oxo-7-[4-(4-phenyl-3-pyridin-3-yl-5-thioxo-4,5-dihydro-[1,2,4]triazol-1-ylmethyl)-piperazin-1-yl]-1,4-dihydro-quinoline-3-carboxylic acid (**11**)

Yield (83%); m.p. 280 °C; IR: 3318 (OH), 3040 (CH aromatic), 2957, 2924 (CH-aliphatic), 1719 (C=O), 1608 (C=N); ^1^H NMR: δ 15.14 (brs. OH, COOH), 8.66–7.41 (m, 1H, aromatic), 5.32 (s, 2H, N-CH_2_-N), 3.83 (m, 1H, CH_2_-CH-CH_2_ cyclopropyl-H), 3.41–3.40 (t, J = 1.44 Hz, 4H, piperazinyl-H), 3.11–3.10 (t, J = 1.72 Hz, 4H, piperazinyl-H), 1.34–1.32 (q, J = 2.16 Hz, 2H, CH_2_-CH-CH_2_ cyclopropyl-H), 1.18 (q, J = 0.84 Hz, 2H, CH_2_-CH-CH_2_cyclopropyl-H); ^13^C NMR: δ 176.81, 170.23, 16.40, 166.40, 149.29, 148.46, 147.66, 145.57, 139.62, 136.53, 135.13, 130.23, 123.92, 122.40, 119.15, 111.52, 107.20, 107.02, 69.39, 50.04, 49.92, 36.32, 19.02, 8.05. C_31_H_28_FN_7_O_3_S: C, 62.30; H, 4.72; N, 16.41, Found: C, 62.14; H, 4.15; N, 16.20.

##### 1-Cyclopropyl-6-fluoro-4-oxo-7-[4-(4-phenyl-3-pyridin-4-yl-5-thioxo-4,5-dihydro-[1,2,4]triazol-1-ylmethyl)-piperazin-1-yl]-1,4-dihydro-quinoline-3-carboxylic acid (**12**)

Yield (91%); m.p. 227 °C; IR: 3321 (OH), 3021(CH aromatic), 2968, 2904 (CH-aliphatic), 1717 (C=O), 1607 (C=N);^1^H NMR: δ 15.06 (brs. OH, COOH), 8.65–7.29 (m, 12H, aromatic), 5.32 (s, 2H, N-CH_2_-N), 3.83–3.81 (m, J = 3.12 Hz, 1H, CH_2_-CH-CH_2_ cyclopropyl-H), 3.41(t, 4H, piperazinyl-H), 3.10 (t, 4H, piperazinyl-H), 1.34–1.32 (q, J = 1.48 Hz, 2H, CH_2_-CH-CH_2_ cyclopropyl-H), 1.18–1.16 (q, J = 2.52 Hz, 2H, CH_2_-CH-CH_2_cyclopropyl-H); ^13^C NMR: δ 178.82, 170.70, 166.35, 150.57, 148.43, 147.57, 145.84, 135.84, 135.13, 133.34, 130.37, 130.00, 125.14, 122.57, 111.53, 107.28, 106.99, 67.53, 50.08, 49.94, 36.30, 19.00, 8.04. C_31_H_28_FN_7_O_3_S: C, 62.30; H, 4.72; N, 16.41, Found: C, 62.16; H, 4.53; N, 16.34.

##### 1-Cyclopropyl-6-fluoro-4-oxo-7-[4-(5-oxo-4,4-diphenyl-2-thioxo-imidazolidin-1-ylmethyl)-piperazin-1-yl]-1,4-dihydro-quinoline-3-carboxylic acid (**13**)

Yield (84%); m.p. 217 °C; IR: 3412 (NH), 3315 (OH), 3031 (CH aromatic), 2981, 2914 (CH-aliphatic), 1721 (C=O), 1601 (C=N); ^1^H NMR: δ 15.07 (brs. OH, COOH), 11.65 (brs. NH, NH imidazolyl), 8.61–7.35 (m, 13H, aromatic), 4.85 (s, 2H, N-CH_2_-N), 3.79–3.77 (m, 1H, CH_2_-CH-CH_2_ cyclopropyl-H), 3.49 (t, 4H, piperazinyl-H), 2.83 (t, 4H, piperazinyl-H), 1.31–1.06 (m, J = 4.48 Hz, 4H, CH_2_-CH-CH_2_ cyclopropyl-H); ^13^C NMR: δ 182.95, 175.24, 166.35, 166.36, 139.51, 138.61, 129.28, 128.91, 121.88, 118.91, 111.54, 107.29, 106.12, 71.74, 62.54, 65.53, 49.84, 36.22, 18.93, 8.82. C_33_H_30_FN_5_O_4_S: C, 64.80; H, 4.94; N, 11.45, Found: C, 64.64; H, 4.77; N, 11.41.

##### 7-[4-(3-Amino-5-oxo-4,4-diphenyl-2-thioxo-imidazolidin-1-ylmethyl)-piperazin-1-yl]-1-cyclopropyl-6-fluoro-4-oxo-1,4-dihydro-quinoline-3-carboxylic acid (**14**)

Yield (79%); m.p. 234 °C; IR: 3391, 3300 (NH_2_), 3259 (OH), 3098 (CH aromatic), 2984, 2912, 2847 (CH-aliphatic), 1708 (C=O), 1601 (C=N); ^1^H NMR: δ 11.81 (brs. OH, COOH), 9.80–9.66 (brs. 2H, NH_2_), 8.41–7.01 (m, 13H, aromatic), 5.14 (s, 2H, N-CH_2_-N), 3.14 (m, 1H, CH_2_-CH-CH_2_ cyclopropyl-H), 2.95–2.74 (t, 4H, piperazinyl-H), 2.26 (t, 4H, piperazinyl-H), 1.04–0.86 (q, 4H, CH_2_-CH-CH_2_ cyclopropyl-H); ^13^C NMR: δ 176.48, 171.24, 166.32, 154.68, 152.31, 151.54, 141.48, 133.11, 131.43, 131.14, 129.39, 129.11, 122.34, 118.41, 117.42, 116.92, 116.69, 67.24, 61.59, 59.34, 54.43, 51.33, 31.31, 23.39, 22.44, 9.11, 8.91; C_33_H_31_FN_6_O_4_S: C, 63.24; H, 4.99; N, 13.41, Found: C, 63.12; H, 4.86; N, 13.37.

##### 7-[4-(2-Cyanoimino-5-oxo-4,4-diphenyl-imidazolidin-1-ylmethyl)-piperazin-1-yl]-1-cyclopropyl-6-fluoro-4-oxo-1,4-dihydro-quinoline-3-carboxylic acid (**15**)

Yield (95%); m.p. 230 °C; IR:3385 (NH), 3330 (OH), 3013 (CH aromatic), 2984, 2901 (CH-aliphatic), 2231 (CN), 1714 (C=O), 1602 (C=N); ^1^H NMR: δ 15.06 (brs. OH, COOH), 11.20 (brs. NH, NH-CN), 8.63–7.40 (m, 13H, aromatic), 4.61 (s, 2H, N-CH_2_-N), 3.37 (m, 1H, CH_2_-CH-CH_2_ cyclopropyl-H), 3.33 (t, 4H, piperazinyl-H), 2.76 (t, 4H, piperazinyl-H), 1.31–1.11 (q, 2H, CH_2_-CH-CH_2_ cyclopropyl-H); ^13^C NMR: δ 194.38, 176.78, 166.31, 148.31, 145.35, 138.65, 129.32, 121.36, 111.55, 106.16, 62.81, 58.11, 49.16, 36.38, 8.85: C_34_H_30_FN_7_O_4_: C, 65.90; H, 4.88; N, 15.82; Found: C, 65.71; H, 4.78; N, 15.77.

##### 7-[4-(5-Benzylidene-2,4-dioxo-thiazolidin-3-ylmethyl)-piperazin-1-yl]-1-cyclo-propyl-6-fluoro-4-oxo-1,4-dihydro-quinoline-3-carboxylic acid (**16**)

Yield (97%); m.p. 237 °C; IR: 3314 (OH), 3017 (CH aromatic), 2984, 2946 (CH-aliphatic), 1711 (C=O), 1603 (C=N); ^1^H NMR: δ 15.10 (brs. OH, COOH), 8.64 (s, 1H, CH=C, CH olefinic) 8.61–7.49 (m, 8H, aromatic), 4.49 (s, 2H, N-CH_2_-N), 3.81–3.77 (m, J = 8.00 Hz, 1H, CH_2_-CH-CH_2_ cyclopropyl-H), 3.46–3.36 (t, 4H, piperazinyl-H), 3.81–2.73 (t, 4H, piperazinyl-H), 1.35–1.32 (q, J = 1.42 Hz, 2H, CH_2_-CH-CH_2_ cyclopropyl-H), 1.19–1.07 (q, J = 2.50 Hz, 2H, CH_2_-CH-CH_2_ cyclopropyl-H); ^13^C NMR: δ 176.34, 168.96, 166.31, 154.63, 152.15, 148.28, 145.58, 139.55, 133.46, 131.81, 130.51, 129.28, 118.91, 111.45, 111.22, 107.21, 106.16, 63.51, 56.53, 50.11, 49.43, 36.25, 18.95, 8.86. C_28_H_25_FN_4_O_5_S: C, 61.30; H, 4.59; N, 10.21, found: C, 61.25; H, 4.52; N, 10.10.

##### 7-{4-[5-(4-Chloro-benzylidene)-2,4-dioxo-thiazolidin-3-ylmethyl]-piperazin-1-yl}-1-cyclopropyl-6-fluoro-4-oxo-1,4-dihydro-quinoline-3-carboxylic acid (**17**)

Yield (96%); m.p. 244 °C; IR: 3290 (OH), 3008 (CH aromatic), 2994, 2944 (CH-aliphatic), 1711 (C=O), 1611 (C=N); δ 15.12 (brs. OH, COOH), 8.64 (s, 1H, CH=C, CH olefinic) 7.91–7.54 (m, 7H, aromatic), 4.70 (s, 2H, N-CH_2_-N), 3.82–3.80 (m, J = 4.00 Hz, 1H, CH_2_-CH-CH_2_ cyclopropyl-H), 3.45–3.34 (t, 4H, piperazinyl-H), 2.82–2.75 (t, 4H, piperazinyl-H), 1.34–1.31 (q, J = 4.00 Hz, 2H, CH_2_-CH-CH_2_ cyclopropyl-H), 1.20–1.07 (q, J = 8.00 Hz, 2H, CH_2_-CH-CH_2_ cyclopropyl-H); ^13^C NMR: δ 176.19, 167.16,166.30, 148.31, 145.53, 139.61, 132.13, 129.88, 119.15, 111.51, 111.21, 107.25, 106.96, 63.62, 59.15, 49.85, 36.28, 18.99, 8.82; C_28_H_24_ClFN_4_O_5_S: C, 57.68; H, 4.15; N, 9.61, found: C, 57.52; H, 4.04; N, 9.53.

##### 1-Cyclopropyl-6-fluoro-7-{4-[5-(4-methoxy-benzylidene)-2,4-dioxo-thiazolidin-3-ylmethyl]-piperazin-1-yl}-4-oxo-1,4-dihydro-quinoline-3-carboxylic acid (**18**)

Yield (92%); m.p. 251 °C; IR: 3349 (OH), 3014 (CH aromatic), 2988, 2908 (CH-aliphatic), 1720 (C=O), 1607 (C=N); ^1^H NMR: δ 15.10 (brs. OH, COOH), 8.64 (s, 1H, CH=C, CH olefinic) 8.62–7.09 (m, 7H, aromatic), 4.68 (s, 2H, N-CH_2_-N), 3.83(s, 3H, OCH_3_), 3.47 (m, 1H, CH_2_-CH-CH_2_ cyclopropyl-H), 3.33–3.09 (t,4H, piperazinyl-H), 3.81–2.72 (t, 4H, piperazinyl-H), 1.35–1.32 (q, 2H, CH_2_-CH-CH_2_ cyclopropyl-H), 1.33–1.05 (q, 2H, CH_2_-CH-CH_2_ cyclopropyl-H); ^13^C NMR: δ 176.78, 169.81, 167.35, 166.33, 166.01 152.16, 148.39, 145.49, 139.54, 133.52, 132.69, 125.94, 119.19, 118.62, 115.45, 111.49, 111.36, 107.36, 106.91, 63.37, 56.54, 55.91, 54.21, 49.84, 36.26, 19.88, 8.82. C_29_H_27_FN_4_O_6_S: C, 60.20; H, 4.70; N, 9.68, found: C, 60.13; H, 4.64; N, 9.53.

##### 1-Cyclopropyl-6-fluoro-7-{4-[5-(2-hydroxy-benzylidene)-2,4-dioxo-thiazolidin-3-ylmethyl]-piperazin-1-yl}-4-oxo-1,4-dihydro-quinoline-3-carboxylic acid (**19**)

Yield (94%); m.p. 244 °C; IR: 3410, 3312 (2OH), 3005 (CH aromatic), 2951, 2910 (CH-aliphatic), 1723 (C=O), 1603 (C=N); ^1^H NMR: δ 15.10 (brs. OH, COOH), 10.51 (brs. OH, PhOH), 8.63 (s, 1H, CH=C, CH olefinic) 8.61–6.92 (m, 7H, aromatic), 4.67 (s, 2H, N-CH_2_-N), 3.81–3.79 (m, J = 4.00 Hz, 1H, CH_2_-CH-CH_2_ cyclopropyl-H), 3.81–3.77 (t, 4H, piperazinyl-H), 2.81–2.74 (t, 4H, piperazinyl-H), 1.33–1.09 (m, 4H, CH_2_-CH-CH_2_cyclopropyl-H); ^13^C NMR: δ 176.73, 169.24, 167.45, 166.32, 157.88, 154.61, 152.13, 148.26, 145.49, 145.39, 139.54, 132.94, 128.41, 128.11, 119.85, 116.67, 111.45, 111.22, 107.22, 106.84, 63.53, 56.51, 50.21,49.81, 46.15, 36.24, 8.88. C_28_H_25_FN_4_O_6_S: C, 59.57; H, 4.46; N, 9.92; Found: C, 59.51; H, 4.40; N, 9.78.

##### 1-Cyclopropyl-7-[4-(1,3-dioxo-1,3-dihydro-isoindol-2-ylmethyl)-piperazin-1-yl]-6-fluoro-4-oxo-1,4-dihydro-quinoline-3-carboxylic acid (**20**)

Yield (89%); m.p. 241 °C; IR: 3341 (OH), 3023 (CH aromatic), 2998, 2812 (CH-aliphatic), 1753, 1721 (2 C=O), 1611 (C=N); ^1^H NMR: δ 15.11 (brs. OH, COOH), 8.64–7.50 (m, 7H, aromatic), 4.56 (s, 2H, N-CH_2_-N), 3.81–3.77 (m, J = 3.12 Hz, 1H, CH_2_-CH-CH_2_ cyclopropyl-H), 3.42 (t, 4H, piperazinyl-H), 2.91 (t, 4H, piperazinyl-H), 1.34–1.29 (q, J = 4.21 Hz, 2H, CH_2_-CH-CH_2_ cyclopropyl-H), 1.19–1.13 (q, J = 4.12 Hz, 2H, CH_2_-CH-CH_2_cyclopropyl-H); ^13^C NMR: δ 176.77, 169.23, 166.38, 154.63, 152.16, 148.25, 145.53, 139.63, 135.83, 134.13,, 131.99, 123.66, 123.33, 118.99, 111.47, 111.29, 107.32, 106.19, 59.39, 54.22, 49.46, 36.23, 7.94. C_26_H_23_FN_4_O_5_: C, 63.67; H, 4.73; N, 11.42, Found: C, 63.51; H, 4.64; N, 11.33.

##### 1-Cyclopropyl-7-[4-(2,5-dioxo-pyrrolidin-1-ylmethyl)-piperazin-1-yl]-6-fluoro-4-oxo-1,4-dihydro-quinoline-3-carboxylic acid (**21**)

Yield (88%); m.p. 214 °C; IR: 3412 (OH), 3007 (CH aromatic), 2998, 2987, 2841, 2828, 2810 (CH-aliphatic),1752, 1701 (2 C=O), 1614 (C=N); ^1^H NMR: δ 15.12 (brs. OH, COOH), 8.63–7.51 (m, 3H, aromatic), 4.34 (s, 2H, N-CH_2_-N), 3.83–3.80 (m, J = 2.02 Hz, 1H, CH_2_-CH-CH_2_ cyclopropyl-H), 3.79–3.77 (t, 2H, CO-CH_2_-CH_2_-C=O), 3.49–3.42 (m, 6H, 4H of piperazinyl-H + CO-CH_2_-CH_2_-C=O), 2.72–2.58 (t, 4H, piperazinyl-H), 1.34–1.32 (q, J = 1.48 Hz, 2H, CH_2_-CH-CH_2_ cyclopropyl-H), 1.17–1.08 (q, J = 2.58 Hz, 2H, CH_2_-CH-CH_2_cyclopropyl-H); ^13^C NMR: δ 179.15, 176.88, 166.32, 154.67, 152.28, 148.28, 145.62, 145.52, 139.64, 119.83, 111.49, 111.36, 107.34, 106.19, 59.52, 56.53, 54.25, 49.87, 36.24, 30.54, 18.93, 8.88.: C_22_H_23_FN_4_O_5_: C, 59.72; H, 5.24; N, 12.66, Found: C, 59.59; H, 5.08; N, 12.49.

### 3.2. Antimicrobial Activity

#### 3.2.1. Organisms and Culture Conditions

The cultures used were collected from the Cairo University’s Microanalytical Centre, Faculty of Science. An updated Kirby-Bauer disc diffusion method was applied for antimicrobial activities of the tests **1–21** compounds [32,33,34,35]. See Appendix B.

#### 3.2.2. Minimum Inhibitory Concentration Assay

On 96-well microtiter plates and 50 mL of fresh bacterial culture of a single McFarland unit overnight, a double serial dilution of each compound (100 mL) in sterile standard saline was prepared for every single source well. Ciprofloxacin antibiotic (5 mg/mL^−1^) and normal saline were included as standard references in each assay [36] (see Appendix B).

#### 3.2.3. Inhibitory Activity Assays on *E. coli* DNA Gyrase and Topoisomerase IV

IC_50_ assay determination was carried out in accordance with the procedures previously stated [37] (see Appendix B).

#### 3.2.4. MTT Assay

MTT Assay was performed to investigate the effect of compounds **2–6** on the viability of mammary epithelial cells (MCF-10A) [38,39] (see Appendix B).

## 4. Conclusions

Several novel heteroaryl-based ciprofloxacin derivatives have been developed. Twenty-one target compounds were synthesized and tested for their in vitro antibacterial activity against bacterial strains of both Gram-positive and Gram-negative using ciprofloxacin as a reference. Most of the compounds examined had evident inhibitory antibacterial activity. Among those compounds, **2**–**6** were the most potent ones. The findings showed that the compounds tested displayed little or poor antifungal activity against *A. flavus* and *C. albicans*. Oxadiazole-based derivatives **4**, **5**, and **6** were found to be the most active and their inhibitory activity against *E. coli* DNA gyrase and Topoisomerase IV was superior to novobiocin with no cytotoxic effects. These compounds, after further optimization, form a new class of antibacterial molecules that target DNA gyrase and topoisomerase IV.

## Data Availability

Not applicable.

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
