# Peer review of "Design, Synthesis, and Antibacterial Screening of Some Novel Heteroaryl-Based Ciprofloxacin Derivatives as DNA Gyrase and Topoisomerase IV Inhibitors"

_pharmaceuticals, 2021, doi:10.3390/ph14050399_

Round 1

Reviewer 1 Report

The presented manuscript edited Design, synthesis, and antibacterial screening of some novel heteroaryl-based ciprofloxacin derivatives as DNA gyrase and topoisomerase IV inhibitors” deals with a current and important topic.

Overall, the manuscript was well organized with well-described. The authors present an analysis of the current state of knowledge from a very innovative point, which makes for me, as a pharmacist, the work is very interesting and valuable.

However, some points need to be clarified before publication. There are few suggestions to improve the manuscript:

  1. FQs belongs to antibiotics that are difficult to biodegrade in the environment - can the authors suppose or carry out computational analyzes of how this aspect will look like in the case of their compounds?
  2. The authors should try to assess the lipophilicity of newly synthesized hydrides using an experimental or computational approach. Especially that the modification of FQs structures can significantly change its properties. Please refer to the actual works in this topic and added relevant information to the manuscript.

To summarizing, the work may be published in Pharmaceutics after minor revision.

Author Response

Reviewer 1

The presented manuscript edited “Design, synthesis, and antibacterial screening of some novel heteroaryl-based ciprofloxacin derivatives as DNA gyrase and topoisomerase IV inhibitors” deals with a current and important topic.

Overall, the manuscript was well organized with well-described. The authors present an analysis of the current state of knowledge from a very innovative point, which makes for me, as a pharmacist, the work is very interesting and valuable.

However, some points need to be clarified before publication. There are few suggestions to improve the manuscript:

  1. FQs belongs to antibiotics that are difficult to biodegrade in the environment - can the authors suppose or carry out computational analyzes of how this aspect will look like in the case of their compounds?

Response: We appreciate the insightful reviewer's feedback and agree with his helpful suggestion. As a result, a drug-likeness profile was determined for all newly synthesized compounds and included in the revised version.

The authors should try to assess the lipophilicity of newly synthesized hydrides using an experimental or computational approach. Especially that the modification of FQs structures can significantly change its properties. Please refer to the actual works in this topic and added relevant information to the manuscript.

Response:

Done as advised.

To summarizing, the work may be published in Pharmaceutics after minor revision.

Reviewer 2 Report

This work is well presented, but there are some key aspects that need to be addressed before this paper is ready to be accepted. I encourage the authors to re-submit this work is a communication once these have been addressed

Major –

The alkenes of ‘IVa-d’ and ‘16-19’ are written incorrectly - what is the isomer ratio? This need to be addressed with 2D NMR experiments to prove the isomers presented.

The numbering system needs to be revised and novel compounds characterised and reported correctly.

The chemistry aspect of this paper needs to be addressed in more detail (also with appropriate references to previous compounds).

The paper needs to be proof-read, there are a lot of errors in here.

Minor -

The fungi data is less interesting, might be better to move this to the Supporting information and reference that the activity was limited but the compounds were tested.

The format of structures is strange, please use chemdraw and make them consistent.

Table 1 needs to be cleaned and formatted correctly

'gram' is written 'g' etc etc, there are lots of examples of these types of issues

References need to be formatted correctly 

Supporting Information - this is not an extensive list of errors:

Compound 19, 18, 17 & 16 needs to be cleaned so that the double bond it properly written

Compound 20, 21 needs to be cleaned so bonds are all equal

'ml' should be written 'mL', 'μl' as 'μL', oC should have a space between oC and the number, mL-1 needs to have the -1 subscripted

p-iodonitrotetrazolium, the 'p' needs to be in italics, MgCl2 needs to have the 2 subscript

- there are many errors like this

Author Response

Reviewer 2

This work is well presented, but there are some key aspects that need to be addressed before this paper is ready to be accepted. I encourage the authors to re-submit this work is a communication once these have been addressed

Major –

  • The alkenes of ‘IVa-d’ and ‘16-19’ are written incorrectly - what is the isomer ratio? This need to be addressed with 2D NMR experiments to prove the isomers presented.

Response:

We would like to thank the Reviewer for this comment. During the work-up procedures for compounds 16-19, we get only one isomer, which precipitated from hot solvent as a single isomer, as verified by TLC (single spot) and spectral results.

The numbering system needs to be revised and novel compounds characterized and reported correctly.

Response

Done as advised

  • The chemistry aspect of this paper needs to be addressed in more detail (also with appropriate references to previous compounds).

Response

Done as advised

  • The paper needs to be proof-read, there are a lot of errors in here.

 Response

The whole manuscript was revised, and all necessary corrections were made

Minor -

  • The fungi data is less interesting, might be better to move this to the Supporting information and reference that the activity was limited but the compounds were tested.

Response: We are thankful to respected reviewer and we agree with his valuable suggestion, but we just want to publish the whole manuscript as one piece due to funding issues.

  • The format of structures is strange, please use chemdraw and make them consistent.

Response

Done as advised

  • Table 1 needs to be cleaned and formatted correctly

Response

Done as advised

  • 'gram' is written 'g' etc etc, there are lots of examples of these types of issues

Response

Done as advised

  • References need to be formatted correctly 

 Response

Done as advised

  • Supporting Information - this is not an extensive list of errors:

Response:

The whole supporting information file was revised, and all necessary corrections were done.

  • Compound 19, 18, 17 & 16 needs to be cleaned so that the double bond it properly written
  • Compound 20, 21 needs to be cleaned so bonds are all equal
  • 'ml' should be written 'mL', 'μl' as 'μL', oC should have a space between oC and the number, mL-1 needs to have the -1 subscripted

Response

All done as advised

14- p-iodonitrotetrazolium, the 'p' needs to be in italics, MgCl2 needs to have the 2 subscripts

- there are many errors like this

Response

Done as advised, moreover the whole manuscript was revised, and we try our best to improve the writing process.

Reviewer 3 Report

The study by Abdelhamid et al reported the synthesis of a small set of heteroaryl derivatives of ciprofloxacin. The authors investigated the antibacterial and antifungal activity of synthesized compounds and found that some of the synthesized exhibited moderate antibacterial activity compared to ciprofloxacin (except for compd 4 and 5 which showed similar activity). However, non of synthesized compounds showed any antifungal activity. Interestingly, the authors found that some of the synthesized compounds inhibited DNA grasse in nano molar range compared to that of novobiocin. This is an interesting study but lack of novelty. Several studies have been reported for hetero-derivatives of ciprofloxacin. Even some of them appear to be more active than the presented derivatives. Additionally, there is no new chemistry has been applied for the synthesis. I think, pharmaceuticals journal seeks for the originality and quality of work, which I do not see in the presented work. I would recommend that the authors direct this work to Molecules journal. Nevertheless, I have some concerns which are listed below:

  • the chemical scheme for the synthesis of compounds does not contain any synthetic details. The authors just draw the compounds above arrows!! the reagents, conditions, yield must be included. Also, the authors draw the heterocyclic as Ia-f (e.g.,) and don't mention what's a-f? In structure I, I see only position to change is Ar? same IIa-f (what'S x and y), IIIa-c, IVa-d, V, VI?
  • from the small set of heteroaryl ciprofloxacin derivatives that has been synthesized and tested, the authors should conclude/draw an overview about the structural features that are beneficial for antibacterial activity/DNA gyrase of this scaffold. Please add something in this direction in the discussion part.
  • What gap in the current scientific landscape on DNA grasse is filled by this work?
  • How the presented inhibitors by the authors are different from the previously known ciprofloxacin derivatives ? it is clear from the results that the activity of the most of synthesized compounds is less than that of ciprofloxacin (except for compound 4 and 5 which showed similar activity)?
  • the antibacterial activity of compounds 4 and 5 seems to be in similar/lower range compared to ciprofloxacin as presented by MIC and inhibition zone experiment, the authors showed discuss how could be the reason for high inhibitory activity toward DNA grasse.
  • C-NMR some peaks are missing for compounds, please carefully check the spectra
  • the introduction part is very short and lack from citing the relevant studies. The authors should discuss more about previously known ciprofloxacin heterocyclic derivatives and their activity and selectivity.
  • since the authors didn't submit the MS as MDPI style, it is so difficult to mention all language mistakes through MS. Please numbering lines so that we can specify.. Nevertheless, the MS needs editing of English language...

Author Response

Reviewer 3

The study by Abdelhamid et al reported the synthesis of a small set of heteroaryl derivatives of ciprofloxacin. The authors investigated the antibacterial and antifungal activity of synthesized compounds and found that some of the synthesized exhibited moderate antibacterial activity compared to ciprofloxacin (except for Compd 4 and 5 which showed similar activity). However, none of synthesized compounds showed any antifungal activity. Interestingly, the authors found that some of the synthesized compounds inhibited DNA gyrase in nano molar range compared to that of novobiocin. This is an interesting study but lack of novelty. Several studies have been reported for hetero-derivatives of ciprofloxacin. Even some of them appear to be more active than the presented derivatives. Additionally, there is no new chemistry has been applied for the synthesis. I think, pharmaceuticals journal seeks for the originality and quality of work, which I do not see in the presented work. I would recommend that the authors direct this work to Molecules journal. Nevertheless, I have some concerns which are listed below:

  • The chemical scheme for the synthesis of compounds does not contain any synthetic details. The authors just draw the compounds above arrows!! the reagents, conditions, yield must be included. Also, the authors draw the heterocyclic as Ia-f (e.g.,) and don't mention what's a-f? In structure I, I see only position to change is Ar? same IIa-f (what'S x and y), IIIa-c, IVa-d, V, VI?

Response

The writers appreciate the reviewer's useful feedback. As a result, scheme 1 for target compound synthesis was revised, and all necessary corrections were made.

  • From the small set of heteroaryl ciprofloxacin derivatives that has been synthesized and tested, the authors should conclude/draw an overview about the structural features that are beneficial for antibacterial activity/DNA gyrase of this scaffold. Please add something in this direction in the discussion part.

Response

We appreciate this comment from the reviewer. Accordingly, we have modified the discussion part in the revised manuscript.

  • What gap in the current scientific landscape on DNA gyrase is filled by this work?

Response

We appreciate this comment from the reviewer. Oxadiazole-based derivatives 4, 5 and 6 were found to be active as antibacterial agents, their inhibitory activity against E. coli DNA gyrase and Topoisomerase IV was superior to novobiocin with no cytotoxic effects. These compounds are currently under led optimization to get a new class of antibacterial molecules that target DNA gyrase and topoisomerase IV.

  • How the presented inhibitors by the authors are different from the previously known ciprofloxacin derivatives? it is clear from the results that the activity of the most of synthesized compounds is less than that of ciprofloxacin (except for compound 4 and 5 which showed similar activity)?

Response

We appreciate this comment from the reviewer. Oxadiazole-based derivatives 4, 5 and 6 were found to be active as antibacterial agents, their inhibitory activity against E. coli DNA gyrase and Topoisomerase IV was superior to novobiocin with no cytotoxic effects. These compounds are currently under led optimization to get a new class of antibacterial molecules that target DNA gyrase and topoisomerase IV.

  • The antibacterial activity of compounds 4 and 5 seems to be in similar/lower range compared to ciprofloxacin as presented by MIC and inhibition zone experiment, the authors showed discuss how could be the reason for high inhibitory activity toward DNA gyrase.

Response: We appreciate the reviewer's time and effort. We also want to point out that the data in our manuscript is merely a straightforward summary of 1-21's antibacterial experiments, which we did honestly as trained medicinal chemists in order to provide readers with a clear picture. Often, instead of using less active references like ampicillin, we use ciprofloxacin (one of the most commonly used antibiotics) as a reference drug.

  • C-NMR some peaks are missing for compounds, please carefully check the spectra

Response:

The whole supplementary file for spectral data was revised and all necessary corrections were made.

  • The introduction part is very short and lack from citing the relevant studies. The authors should discuss more about previously known ciprofloxacin heterocyclic derivatives and their activity and selectivity.

Response

We appreciate this comment from the reviewer. Accordingly, we have modified the introductory part in the revised manuscript

  • Since the authors didn't submit the MS as MDPI style, it is so difficult to mention all language mistakes through MS. Please numbering lines so that we can specify. Nevertheless, the MS needs editing of English language...

Response:

The whole manuscript was revised, and we try our best to improve the writing process.

Reviewer 4 Report

The authors described a new series of ciprofloxacin hybrids with antibacterial activity and inhibitory potency on E. coli DNA gyrase in the nanomolar range and on topoisomerase IV in the micromolar range. Comments and suggested improvements to the manuscript are suggested below.

- Abstract: bacterial strains please italicise.

- Introduction: authors should mention side effects and problems of fluoroquionolones with development of resistance.

- Through the manuscript: write Staphylococcus aureus first, then you can write S. aureus. This also applies to other bacterial strains throughout the manuscript.

- The authors should provide the explanation why these molecules are hybrid molecules? They seem to be rather mere derivatives of the cyprofloxacin molecule.

- ml should be written mL

- The oxadiazole derivatives were the most potent. I would suggest performing cytotoxicity measurements on at least one cell line for these molecules.

- A discussion of structure-activity relationships should be given

- It should be explained what the advantages of these molecules are compared to ciprofloxacin.

Author Response

Reviewer 4

The authors described a new series of ciprofloxacin hybrids with antibacterial activity and inhibitory potency on E. coli DNA gyrase in the nanomolar range and on topoisomerase IV in the micromolar range. Comments and suggested improvements to the manuscript are suggested below.

1- Abstract: bacterial strains please italicize.

Response

Done as advised

2- Introduction: authors should mention side effects and problems of fluoroquinolones with development of resistance.

Response

We appreciate this comment from the reviewer. Accordingly, we have modified the introductory part in the revised manuscript per the reviewer request.

3- Through the manuscript: write Staphylococcus aureus first, then you can write S. aureus. This also applies to other bacterial strains throughout the manuscript.

Response

Done as advised

4- The authors should provide the explanation why these molecules are hybrid molecules? They seem to be rather mere derivatives of the ciprofloxacin molecule.

Response

Currently, molecular hybridization is considered a promising approach where two or more biologically active scaffolds are fused to generate novel agents against the desired drug target. With a molecular hybridization approach, we synthesized 21 azole derivatives, which were then converted to fluoroquinolone hybrids by the three-component Mannich reaction.

5- ml should be written mL

Response

Done as advised

6- The oxadiazole derivatives were the most potent. I would suggest performing cytotoxicity measurements on at least one cell line for these molecules.

Response

We would like to thank the reviewer for his helpful input. Accordingly, the cell viability assay of oxadiazole derivatives 2-6 were performed where no cytotoxic effects were reported for the tested compounds and the cells' viability for most of the compounds tested was greater than 85% at 50 μM".

7- A discussion of structure-activity relationships should be given

Response:

Done as advised

8- It should be explained what the advantages of these molecules are compared to ciprofloxacin.

Response

We appreciate this comment from the reviewer. Oxadiazole-based derivatives 4, 5 and 6 were found to be active as antibacterial agents, their inhibitory activity against E. coli DNA gyrase and Topoisomerase IV was superior to novobiocin with no cytotoxic effects. These compounds are currently under led optimization to get a new class of antibacterial molecules that target DNA gyrase and topoisomerase IV.

Round 2

Reviewer 2 Report

The authors have described a new series of ciprofloxacin hybrids with antibacterial activity. I can see that the authors have put efforts in to fix this manuscript, but issues still remain.

Major -

The bond angles and drawing in Scheme 1 is very poor. The 180 degree aldehyde, the alkene (again) drawn elongated and may other issues all related to the addition on the ciprofloxacin sub-structure. This MUST be fixed as this is currently far from publication standard. This is throughout the manuscript (eg Table 2). Do a structural clean up on all compounds.

Looking specifically at TH-3 (cmpd 18) I can almost see other isomer at 176.18 where there are two peaks corresponding to the ketone in the hydantoin scaffold. The authors need to run an NOE experiment to be sure they have just one isomer and clearly demonstrate this.

The Fungi data need to be moved to the supporting information. They can be referenced in the text, but they are inactive and not relevant to the main story. The funders will not mind if this is referenced to the supporting information. It is still useful to know and is in the public domain.

The numbering system in this manuscript is very hard to follow, why not just 1, 2, 3... including the reagents and the compounds from the second fig.1

Minor -

'VIIa-F' should be 'VIIa-f'

The figure numbers are not accurate.

The compounds need to be correctly numbered in the supporting information.

AAM-3 in the supporting information has a strange scale for the NMR, this needs to be fixed.

Author Response

Comments and Suggestions for Authors

The authors have described a new series of ciprofloxacin hybrids with antibacterial activity. I can see that the authors have put efforts in to fix this manuscript, but issues still remain.

Major -

The bond angles and drawing in Scheme 1 is very poor. The 180 degree aldehyde, the alkene (again) drawn elongated and may other issues all related to the addition on the ciprofloxacin sub-structure. This MUST be fixed as this is currently far from publication standard. This is throughout the manuscript (eg Table 2). Do a structural clean up on all compounds.

Done as advised

Looking specifically at TH-3 (cmpd 18) I can almost see other isomer at 176.18 where there are two peaks corresponding to the ketone in the hydantoin scaffold. The authors need to run an NOE experiment to be sure they have just one isomer and clearly demonstrate this.

The presence of isomers can be easily detected by NMR signal duplication, which did not occur in our case.

The Fungi data need to be moved to the supporting information. They can be referenced in the text, but they are inactive and not relevant to the main story. The funders will not mind if this is referenced to the supporting information. It is still useful to know and is in the public domain.

Although we agree with our esteemed reviewer, all co-authors request that the fungal section be included in the main manuscript.

The numbering system in this manuscript is very hard to follow, why not just 1, 2, 3... including the reagents and the compounds from the second fig.1

All new compounds are numbered 1-21, but for previously reported compounds, we use I, II,...

Minor -

'VIIa-F' should be 'VIIa-f'

Done as advised

The figure numbers are not accurate.

Corrected

The compounds need to be correctly numbered in the supporting information.

The correct number is included in the title of the figure.

AAM-3 in the supporting information has a strange scale for the NMR, this needs to be fixed.

 For the right figure and scale, please see page 2 in supporting information

Reviewer 3 Report

The authors have not cover all my comments. Please, consider the following points:

1- The antibacterial activity of compounds 4 and 5 seems to be in similar/lower range compared to ciprofloxacin as presented by MIC and inhibition zone experiment, the authors should discuss how could be the reason for high inhibitory activity toward DNA gyrase.

Comment: the authors did not discuss this point.

2- How the presented inhibitors by the authors are different from the previously known ciprofloxacin derivatives? it is clear from the results that the activity of the most of synthesized compounds is less than that of ciprofloxacin (except for compound 4 and 5 which showed similar activity)?

Comment: the authors did mot cover this point.

3- From the small set of heteroaryl ciprofloxacin derivatives that has been synthesized and tested, the authors should conclude/draw an overview about the structural features that are beneficial for antibacterial activity/DNA gyrase of this scaffold. Please add something in this direction in the discussion part.

Comment: the authors did not mention anything in this direction in the discussion part.

4- The chemical scheme for the synthesis of compounds does not contain any synthetic details. The authors just draw the compounds above arrows!! the reagents, conditions, yield must be included. 

Comment: the authors did not include the conditions, yield...

5- the MS still needs extensive english editing.

Author Response

Comments and Suggestions for Authors

The authors have not cover all my comments. Please, consider the following points:

The antibacterial activity of compounds 4 and 5 seems to be in similar/lower range compared to ciprofloxacin as presented by MIC and inhibition zone experiment, the authors should discuss how could be the reason for high inhibitory activity toward DNA gyrase.

They are two different studies with two different references; for antibacterial activity as MIC or inhibition zone, we use one of the most active reference drugs, ciprofloxacin; for mechanistic research, we use the reported reference (novobiocin) for this assay.

Comment: the authors did not discuss this point.

2- How the presented inhibitors by the authors are different from the previously known ciprofloxacin derivatives? it is clear from the results that the activity of the most of synthesized compounds is less than that of ciprofloxacin (except for compound 4 and 5 which showed similar activity)?

Comment: the authors did mot cover this point.

This is a preliminary study for led identification, and we believe the findings are promising for future development.

3- From the small set of heteroaryl ciprofloxacin derivatives that has been synthesized and tested, the authors should conclude/draw an overview about the structural features that are beneficial for antibacterial activity/DNA gyrase of this scaffold. Please add something in this direction in the discussion part.

Comment: the authors did not mention anything in this direction in the discussion part.

The following paragraph has already been added to section 2.2.2. (The nature of aromatic substitution in the oxadiazole moiety tends to be correlated with higher antibacterial effects and the activity increased with (Ar) in the order of 2-pyridyl ≥ 3-pyridyl > 4-pyridyl > 2-Cl-Ph > 2-naphthyl.)

4- The chemical scheme for the synthesis of compounds does not contain any synthetic details. The authors just draw the compounds above arrows!! the reagents, conditions, yield must be included. 

Comment: the authors did not include the conditions, yield...

All of this information has already been added to the discussion of chemistry section.

5- the MS still needs extensive english editing.

We try our best to improve the writing process

Reviewer 4 Report

The authors have taken all comments into account and improved the manuscript accordingly. I therefore recommend that the article be accepted for publication.

Author Response

thanks to accepting our manusript

Round 3

Reviewer 2 Report

The authors have described a new series of ciprofloxacin hybrids with antibacterial activity. I can see that the authors have put efforts in to fix this manuscript, but issues still remain.

Major -

Have a look at compounds 16-19 and the attached highlighted spectra of the 13C of 16, I can almost see other isomer at 176.14 and other parts of the spectra where there are duplicate peaks corresponding two isomers. The authors need to run an NOE and associated NMR experiments on 16-19 to fully characterize these compounds and clearly demonstrate this a single isomer or the mixture. The presence of isomers can be easily detected by NMR through signal duplication and then further NMR is required to characterize this mixture. There multiple unlabeled peaks in the carbon. The onus is on the authors to produce evidence to the counter by zooming in on each of the analogs 16, 17, 18 and 19 or do the additional NMR experiments. This needs to be addressed before the paper can be accepted. This is a question of scientific soundness and at the moment I believe there is a mixture of isomers in 16-19 that the authors are passing off as single compounds. If this is not the case and it is just bad picture quality (on all four spectra), the authors should show evidence (zoomed in spectra and/or NOE/HMQC/HMBC etc) to counter this point.

The Fungi data and the numbering system are editorial choices, I will leave that to the discretion of the academic editor.

Minor -

The scheme can still be improved and the bond angles cleaned, the authors have made efforts but this still not really fixed.

Author Response

Have a look at compounds 16-19 and the attached highlighted spectra of the 13C of 16, I can almost see other isomer at 176.14 and other parts of the spectra where there are duplicate peaks corresponding two isomers. The authors need to run an NOE and associated NMR experiments on 16-19 to fully characterize these compounds and clearly demonstrate this a single isomer or the mixture. The presence of isomers can be easily detected by NMR through signal duplication and then further NMR is required to characterize this mixture. There multiple unlabeled peaks in the carbon. The onus is on the authors to produce evidence to the counter by zooming in on each of the analogs 16, 17, 18 and 19 or do the additional NMR experiments. This needs to be addressed before the paper can be accepted. This is a question of scientific soundness and at the moment I believe there is a mixture of isomers in 16-19 that the authors are passing off as single compounds. If this is not the case and it is just bad picture quality (on all four spectra), the authors should show evidence (zoomed in spectra and/or NOE/HMQC/HMBC etc) to counter this point.

Response:

According to their 1H NMR, our products 16-19 do not exist in an isomeric form. Furthermore, these items appear as a singlet spot on TLC plates. The presence of several peaks in the 13C NMR spectrum is due to the presence of a fluorine atom, which can cause this duplication, i.e., there is Jc-f. As a result, the NOE study is unnecessary due to the lack of any nearby hydrogen atoms that could overlap with adjacent phenyl protons or H. please see the following manuscript (Alexander F. Khlebnikov, Mikhail S. Novikov and Amer A. Amer; Generation and cycloadditions of azirinium difluoromethanides—strained azomethine ylides; Tetrahedron Letters 43 (2002) 8523–8525).

Minor -

The scheme can still be improved, and the bond angles cleaned, the authors have made efforts but this still not really fixed.

Response:

Already done

Reviewer 3 Report

The authors have covered most of the points raised and the MS has been improved. However, I still can see that the MS requires:

Extensive editing of English language and style. I recommend that they use the service offered by MDPI.

The authors did not discuss in the MS How the presented inhibitors by the authors are different from the previously known ciprofloxacin derivatives?

The MS still contains inappropiate self-citations, at least 5 citations for Antar A. Abdelhamid, 7 citations for Bahaa G. M. Youssif, 5 citations for  Marzouk, which means that the authors have almost 17 self-citaions for their articles in this MS, which is totally not-acceptable.

the references part still contains many mistakes, please carefully check it and only use the style of the journal in writing the references.

Author Response

Comments and Suggestions for Authors

The authors have covered most of the points raised and the MS has been improved. However, I still can see that the MS requires:

Extensive editing of English language and style. I recommend that they use the service offered by MDPI.

Response

We tried our hardest, and we believe the manuscript was suitable in terms of English language and style.

The authors did not discuss in the MS How the presented inhibitors by the authors are different from the previously known ciprofloxacin derivatives?

Response

Please read first paragraph in the introductory part (Bacterial infection remains a significant threat to human life due to its increasing resistance to conventional antibiotics, which is a growing public health concern. As a result, there is a critical need to create new antimicrobial agents with potent anti-drug-resistant microorganism activity [1]. That is why antimicrobial agent investigations are so critical and should always be up-to-date). Furthermore, these new derivatives could serve as a good starting point for further modification and production of other more potent targets.

The MS still contains inappropiate self-citations, at least 5 citations for Antar A. Abdelhamid, 7 citations for Bahaa G. M. Youssif, 5 citations for  Marzouk, which means that the authors have almost 17 self-citaions for their articles in this MS, which is totally not-acceptable.

Response

All of the references listed serve the manuscript, whether they are self-ciation or not.

the references part still contains many mistakes, please carefully check it and only use the style of the journal in writing the references.

Response:

All references have been reviewed, and all measures have been taken.

Round 4

Reviewer 2 Report

The authors have made some efforts to resolve the issues raised, but I am not satisfied with the NMRs of 16-19 and the wider NMR reporting in this work. I believe there are 2 isomers present in 16-19 and wider issues with the NMR tabulations (fluorine couplings potentially ignored). The fluorine coupling for direct C-F attachment is typically between 175-275 Hz and the other carbons ranging anything from 2-25 Hz. The coupling should be correctly reported and not just ignored if that is the case. The authors unwillingness to engage on these issues is very unfortunate.

Author Response

In C-NMR for compound 16

It is clear that the two signals at 154.63, 152.15 produced from fluorine coupling for direct C-F

i.e 1JC-F = 250 Hz as shown in the following fig 

111.45, 111.22 due to 2JC-F = 23 Hz

Compounds 16-19 were prepared by the reaction of ciprofloxacin with formaldehyde and 5-arylidenethiazolidine-2,4-dione, which presence in z configuration 

The isomer with Z configuration has little steric hindrance and is thermodynamically more stable, and the 1H NMR shift for the methine hydrogen is also consistent with the Z configuration isomer.1–2

Ben-Yong Yang* and De-Hong Yang, JOURNAL OF CHEMICAL RESEARCH 2011, 238–239

1- J. Dolezel, P. Hirsova, V. Opletalova, J. Dohnal, V. Marcela, J. Kunes and J. Jampilek, Molecules, 2009, 14, 4197.

2- Y. Momose, K. Meguro, H. Ikeda, C. Hatanaka, S. Oi and T. Sohda, Chem. Pharm. Bull., 1991, 39, 1440.
